

# In the social amoeba *Dictyostelium discoideum*, shortened stalks may limit obligate cheater success even when exploitable partners are available

James Medina*, Tyler Larsen*, David C. Queller and Joan E. Strassmann

Department of Biology, Washington University in St. Louis, St. Louis, Missouri, United States
* These authors contributed equally to this work.

## ABSTRACT

Cooperation is widespread across life, but its existence can be threatened by exploitation. The rise of obligate social cheaters that are incapable of contributing to a necessary cooperative function can lead to the loss of that function. In the social amoeba *Dictyostelium discoideum*, obligate social cheaters cannot form dead stalk cells and in chimeras instead form living spore cells. This gives them a competitive advantage within chimeras. However, obligate cheaters of this kind have thus far not been found in nature, probably because they are often enough in clonal populations that they need to retain the ability to produce stalks. In this study we discovered an additional cost to obligate cheaters. Even when there are wild-type cells to parasitize, the chimeric fruiting bodies that result have shorter stalks and these are disadvantaged in spore dispersal. The inability of obligate cheaters to form fruiting bodies when they are on their own combined with the lower functionality of fruiting bodies when they are not represent limits on obligate social cheating as a strategy.

# INTRODUCTION

Cooperative behavior is common in nature, but cooperators can be vulnerable to cheaters who gain the benefits of cooperation without paying the costs (*Bourke, 2013*). In order for cooperation to persist, conflict between cooperators and cheaters must be mitigated (*Smith & Szathmary, 1997*; *Michod, 2000*; *Queller & Strassmann, 2018*). Cheating can be reduced or eliminated by natural selection if the benefits of cooperation preferentially go to relatives because relatives are likely to share the gene or genes underlying the cooperative behavior. This is called kin selection, and more generally inclusive fitness theory. Individuals maximize their inclusive fitness, which includes their personal fitness as well as their effects on the fitness of their relatives, modified by how closely related they are (*Hamilton, 1963*, *1964*; *Queller, 1992*; *Frank, 1998*; *Grafen, 2006*).

The social amoeba *Dictyostelium discoideum* can form a multicellular fruiting body that requires altruistic action by a subset of cells. In the wild, individual amoebas live in soil and leaf litter where they prey upon bacteria. When starved, they aggregate, develop into a

Corresponding author
Tyler Larsen, tjlarsen@wustl.edu

multicellular slug, and migrate to a new location where they form a fruiting body. In this multicellular structure most cells mature into spores, but about 20% of cells altruistically sacrifice their lives to form a dead stalk to lift the other cells up to a few millimeters above the soil (*Kessin, 2001*). Altruism can evolve by kin selection if the spores are genetic relatives of stalk cells, provided the benefits of making a stalk are high enough (*Strassmann, Zhu & Queller, 2000*).

Given that it is costly, why become stalk at all? The benefit of making a fruiting body seems to be to enhance dispersal, because experiments show fruiting bodies with their stalks destroyed are dispersed less well by a model insect vector *Drosophila melanogaster* than those with intact stalks (*Smith, Queller & Strassmann, 2014*).

Social amoeba aggregations can form between unrelated genotypes (*Strassmann, Zhu & Queller, 2000*) or even different species (*Jack et al., 2008*), setting the stage for social conflict between lineages over which cells become spores and which become stalk. In chimeric aggregations, natural selection should favor genotypes that preferentially become spores and place the burden of stalk-building on other genotypes. This conflict can be mitigated through high relatedness between cells within an aggregate (*Gilbert et al., 2007*; *Kuzdzal-Fick et al., 2011*; *Inglis et al., 2017*), as well as by other mechanisms such as pleiotropy (*Foster et al., 2004*; *Belcher et al., 2022*) and a lottery-like role assignment system based on the cell cycle and nutrition (*Strassmann & Queller, 2011*). In *D. discoideum*, the high relatedness necessary for preserving cooperation can be generated by active processes like kin discrimination (*Ostrowski et al., 2008*; *Benabentos et al., 2009*; *Gilbert, Strassmann & Queller, 2012*; *Strassmann, 2016*), or by passive processes like spatial population growth and fine-scale population structure (*Buttery et al., 2012*; *Smith, Strassmann & Queller, 2016*).

There are multiple ways for an amoeba to cheat (*Travisano & Velicer, 2004*; *Santorelli et al., 2008*; *Buttery et al., 2009*; *Strassmann & Queller, 2011*; *Medina, 2019*). Facultative cheaters can change their behavior in different social contexts, overrepresenting themselves in the spores when in chimeras but retaining the ability to make functional fruiting bodies when growing clonally. In *D. discoideum* the mutants $chtB^-$ and $chtC^-$ are examples of this strategy (*Khare & Shaulsky, 2010*; *Santorelli et al., 2013*). In addition, facultative cheaters may tailor their investment into stalk production based on their relatedness to other cells within a chimera (*Madgwick et al., 2018*). Fixed cheaters always allocate the same amount to spores and can be overrepresented in the spores if their fixed strategy happens to be to give more to spores than their social partner does. Allocation to spore *versus* stalk varies in nature so this may be common (*Votaw & Ostrowski, 2017*). Social parasites, or obligate social cheaters, cannot make fruiting bodies on their own and tend to become spores in chimera such as the mutant $chtA^-$ (*Ennis et al., 2000*). Unlike the other categories of cheating, the spread of obligate social cheaters can threaten cooperation itself.

Key to the success or failure of the obligate cheating strategy in the *D. discoideum* system is the relatedness between cells within fruiting bodies, which captures the likelihood of a cheater existing within a chimera with other cell lineages it can exploit. Relatedness within *D. discoideum* fruiting bodies tends to be high in natural environments, which should tend

to magnify the costs of obligate cheating and minimize its benefits (*Gilbert et al., 2007*). Accordingly, no obligate social cheaters have been isolated from naturally occurring fruiting bodies despite extensive sampling (*Gilbert et al., 2007*; *Votaw & Ostrowski, 2017*). In contrast, when relatedness is experimentally lowered in the lab, obligate social cheaters evolve readily (*Ennis et al., 2000*; *Kuzdzal-Fick et al., 2011*; *Inglis et al., 2017*) and can drive the evolution of cheating-resistance within altruistic lineages (*Khare & Shaulsky, 2010*; *Levin et al., 2015*).

In this study, we observed evidence of another cost experienced by obligate cheaters that may further contextualize their rarity in nature. While relatedness within fruiting bodies is often high in nature, chimeras do form (*Gilbert et al., 2007*). Even within chimeric fruiting bodies, however, obligate cheaters may pay a cost if their presence compromises the fruiting bodies' functionality. We reasoned that by not contributing to stalk production, obligate cheaters might impose a burden on a chimeric aggregate, even if it also contained functional cooperative lineages. If exploited lineages do not compensate for the presence of cheaters by allocating more resources to stalk production than they would within a clonal fruiting body, then the resultant chimeric fruiting body is likely to be shorter. This may in turn reduce its dispersal functionality, reducing the likelihood that all lineages within the chimera are dispersed as often or as far. The impact of social behavior on dispersal potential *via* the height of fruiting bodies, rather than their presence or absence, has not yet been experimentally tested.

Here we examine the effect of social conflict between a wild clone of *D. discoideum*, NC28.1, and an obligate, non-fruiting social cheater previously evolved from NC28.1 called EC2 (*Inglis et al., 2017*). Like *fbxA⁻* (*Ennis et al., 2003*), EC2 is an obligate social cheater that cannot produce functional fruiting bodies when grown clonally, but can gain an advantage when cocultured with functional, altruistic strains like NC28.1. We mixed EC2 and NC28.1 at different frequencies to produce chimeric fruiting bodies, then measured the fruiting bodies' heights. We tested whether the cheater strain EC2 would impose costs on chimeric fruiting bodies that may represent drawbacks of the cheating strategy within *D. discoideum*. We tested if fruiting bodies containing higher frequencies of cheaters would be shorter because the cheaters do not contribute to stalk production.

## METHODS

Portions of this text have previously been published as part of a preprint (*Medina et al., 2023*).

### Strains and culture conditions

To prepare food bacteria for *D. discoideum* clones to prey upon, we first spread non-pathogenic *K. pneumoniae* KpGe (Dicty Stock Center, http://dictybase.org/) from stocks frozen in 80% KK2 (2.25 g $KH_2PO_4$ (Sigma-Aldrich, St. Louis, MO, USA) and 0.67 g $K_2HPO_4$ (Fisher Scientific) per liter) and 20% glycerol on an SM/5 agar media (2 g glucose (Thermo Fisher Scientific), 2 g yeast extract (Oxoid), 0.2 g $MgCl_2$ (Thermo Fisher Scientific), 1.9 g $KHPO_4$ (Sigma-Aldrich, St. Louis, MO, USA), 1 g $K_2HPO_5$ (Thermo Fisher Scientific), and 15 g agar (Thermo Fisher Scientific) per liter) and allowed the
bacteria to grow at room temperature until colonies appeared. We picked a single colony with a sterile loop, spread it on a new SM/5 plate, and allowed the bacteria to proliferate. We collected these bacteria into KK2 with a sterile loop and diluted them to 1.5 $OD_{600}$ in KK2 (~$5 \times 10^8$ cells, measured with an Eppendorf BioPhotometer). We used these bacteria as food for amoebas in our experiment and repeated this process anew for each of the three replicate experiments.

To grow NC28.1, the wild-type ancestor *D. discoideum* clone, from freezer stocks for use in our experiments we added spores frozen in 80% KK2 and 20% glycerol to 200 μl of 1.5 $OD_{600}$ *K. pneumoniae* suspension. We spread the mix of spores and bacteria on SM/5 plates with a sterile glass spreader, then incubated the plates at room temperature for 7 days under constant overhead light until the social cycle was complete and fruiting bodies had formed. We repeated this process for each of the three replicate experiments.

EC2 (also called EC28.2 in *Inglis et al. (2017)*) was selected as an obligate cheater. It is the result of an experimental evolution experiment that used unstructured growth and dispersal to evolve a cooperative wild strain called NC28.1 into non-fruiting cheaters. Past work measuring the degree of cheating (the overrepresentation of spores within chimeric fruiting bodies) suggests that EC2 is a particularly effective cheater, and its recent divergence from a wild isolate should render its behavior more natural than cheaters derived from extensively lab-adapted strains. To grow EC2, the RFP-labelled obligate social cheater, from freezer stocks for use in our experiments, we added amoebas frozen in HL5 (5 g proteose peptone, 5 g thiotone E peptone, 10 g glucose, 5 g yeast extract, 0.35 g Na2HPO4 * 7H2O, 0.35 g KH2PO4 per liter) to 10% DMSO to 200 μl of 1.5 $OD_{600}$ *K. pneumoniae* suspension. We spread the mix of amoebas and bacteria on an SM/5 plate with a sterile glass spreader, then incubated the plate at room temperature for 24–48 h until starving EC2 amoebas began aggregating. We then used a sterile loop to transfer a sample to a new plate containing fresh *K. pneumoniae* for them to prey upon. These were allowed to grow for 24–48 h until a vegetative front of amoebas had formed. We collected these amoebas with a sterile loop into ice-cold KK2 (see "Experimental procedures") and ensured that the amoebas we used were clonal by plating 10 SM/5 plates with about 10 amoebas each, then picking a single clonal plaque originating from a single amoeba. We repeated this process for each of the three replicate experiments.

## Experimental procedures

In order to obtain cells of both *D. discoideum* clones for experimental mixing, we plated amoebas (EC2) or spores (NC28.1) previously grown from freezer stocks as described above on separate SM/5 agar plates with 200 μl of 1.5 $OD_{600}$ *K. pneumoniae* suspension.

We collected amoebas to make the mixtures by pouring ice-cold KK2 onto the plates, mixing them into suspension with a gloved fingertip, then collecting and centrifuging the mixture at 10 °C for 3 min at 1,300 rpm in order to pellet the amoebas and leave *K. pneumoniae* in solution. We decanted the pellets, resuspended them in KK2, and measured their density with a hemacytometer before making the mixtures. For each treatment, we mixed 200 μl of fresh *K. pneumoniae* suspension with a total of $2 \times 10^5$ amoebas then spread the solution evenly with an ethanol-sterilized glass spreader on an

SM/5 agar plate. We made mixtures of EC2 and NC28.1 with various initial frequencies of EC2 (0.0, 0.1, 0.3, 0.5, 0.7, 0.9, and 1.0). We repeated this experiment three times, each on a separate day.

We collected fruiting bodies after 1 week at room temperature under constant overhead light to allow fruiting bodies to fully develop. On each plate, we selected three fruiting bodies at random to represent three independent data points. To do this, we placed a plate of fruiting bodies over a grid of 1 cm by 1 cm squares. We selected three squares at random using a random number generator and marked each plate at the centers of each of the squares. We then individually collected the closest intact (not collapsed) fruiting body to each mark with fine tweezers. For each, we pressed the sorus, which contains the spores, against the side of a tube containing 100 µl of KK2 to dislodge the spores, then laid the stalk on a glass microscope slide. After three fruiting bodies were collected from a single plate, the stalks were covered with a cover slip and sealed with nail polish for later imaging. Stalk length was individually recorded by imaging picked stalks under a Leica S8AP0 dissecting microscope with Leica application suite software v4.1 using the "draw line" tool.

### Analysis

We excluded several data points from the analysis for which we could not accurately measure stalk height due to damage incurred during collection.

In order to test whether increasing cheater frequency yields shorter fruiting bodies, we used a linear mixed-effects model with the function lme in the nlme package in R version 4.2.1 (*R Core Team, 2022*) with stalk height as the response variable, the initial cheater frequency as a fixed effect, the total number of spores per sorus as a fixed effect, and the day of the experiment as a random effect (stalk height ~ initial cheater frequency + total spores + 1|day). In this model, we included total number of spores as a fixed effect in case fruiting body height could vary due to random variation in the size of aggregates that form across the plate.

We then compared this model with one lacking the random effect of day (stalk height ~ initial cheater frequency + total spores) using the *anova* function in base R. The two models were not significantly different ($p = 0.27$, with day: AIC = 105.67, without day: AIC = 104.89), so we proceeded with the simpler model without the effect of day. We then further simplified the model by removing the effect of total number of spores because it did not significantly affect stalk height ($p = 0.20$).

### RESULTS

We mixed a wild type *D. discoideum* and an obligate social cheater descendent at various frequencies, allowed them to fruit, and then measured the height of the resultant chimeric fruiting bodies. We found that initial cheater proportion significantly predicted stalk height (linear model, DF = 48, $t = -6.17$, $p = 1.37e-07$). The slope of this relationship was negative (−1.34, SE = 0.22), indicating that for each increase of 10% EC2 in the initial chimeric mix, the resulting fruiting body would be 0.134 mm shorter (Fig. 1). At high cheater frequencies some of the chosen fruiting bodies did not develop and were therefore scored as having zero stalk height. We reanalyzed the data excluding these aggregates.

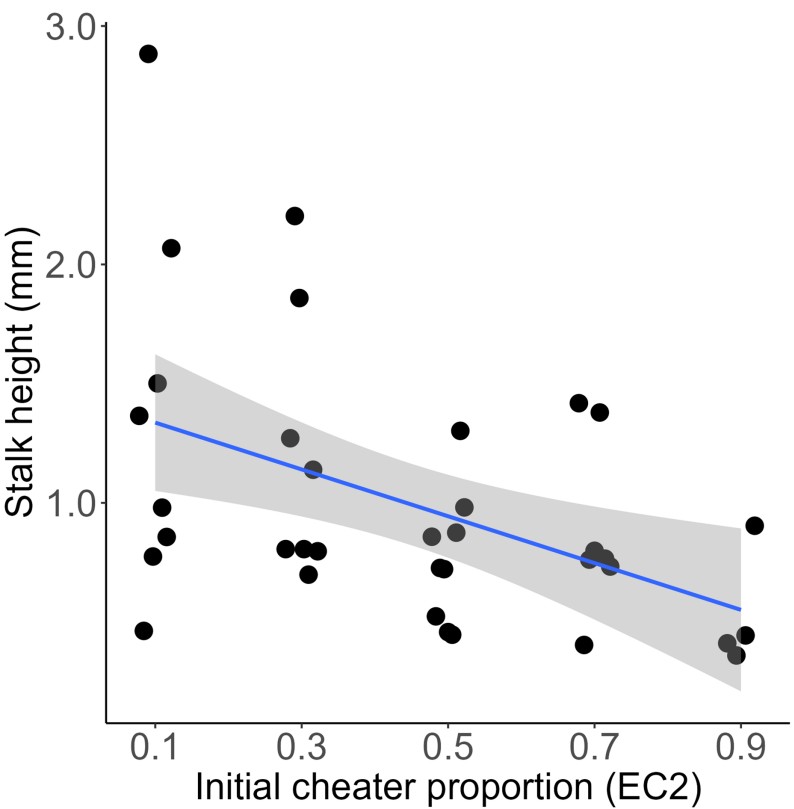

**Figure 1 Chimeras containing higher proportions of obligate cheaters produce significantly shorter stalks.** Each point represents measurement of a single fruiting body. Initial cheater proportion predicts stalk height (linear model, DF = 48, t = −6.17, $p$ = 1.37e-07). Regression line is y = −1.34x + 1.6. R-squared = 0.47. Shaded area is 95% CI.               

Initial cheater proportion still significantly predicted stalk height (linear model, DF = 43, t = −4.83, $p$ = 1.75e-05), and the slope of the relationship remained negative (−1.22, SE = 0.25). Total cell number within each chimera did not significantly affect stalk height of the resulting fruiting bodies (linear model, DF = 34, t = 0.391, $p$ = 0.698) (Fig. S1).

## DISCUSSION

Cooperation is common in microbes, but populations of cooperators can be invaded by cheaters that benefit from cooperation without paying the cost (*West et al., 2007*; *Ghoul, Griffin & West, 2014*). In the extreme case of obligate social cheaters, which are incapable of cooperating entirely, conflict between cheaters and cooperators can destabilize cooperative traits. An obligate cheater that always enjoys a selective advantage over cooperators should rise in frequency within a population until there are too few cooperators for cooperation to occur at all. This sort of takeover and destruction of cooperation by obligate cheaters is readily observed in populations of the social amoeba *Dictyostelium discoideum* evolved in the laboratory (*Kuzdzal-Fick et al., 2011*). However, despite the apparent potential of the strategy, obligate cheaters have never been observed in *D. discoideum* populations in nature.

One potential explanation is that obligate cheaters may not necessarily have an advantage over cooperators. Previous studies in *D. discoideum* found that an obligate social cheater mutant called *fbxA⁻* gains a benefit within chimeras, but is limited by poor spore production when it does not have other clones to exploit (*Ennis et al., 2000*). This cost is likely to be important in nature where relatedness within fruiting bodies is known to be high (*Gilbert et al., 2007*), and so any obligate cheaters like *fbxA⁻* that happen to arise are likely to frequently find themselves alone.

In this study, we explored an additional cost that may help explain the apparent lack of success of obligate cheaters in *D. discoideum* even under lower relatedness conditions. We hypothesized that even if relatedness was low enough that obligate cheaters could consistently aggregate with cooperators that they could exploit, the cheaters' inability to contribute to stalk production would lead to shorter, less functional fruiting bodies. To test this idea, we combined a naturally occurring *D. discoideum* clone with a previously evolved obligate cheater descendent at multiple starting frequencies, allowed them to form chimeric fruiting bodies, and measured the height of the resulting chimeric fruiting bodies.

As we predicted, we found that chimeric fruiting bodies have significantly shorter stalks than fruiting bodies produced by clonal aggregates (Fig. 1). Stalk production is a complex and costly process that is likely to be important for the fitness of *D. discoideum* and its dictyostelid relatives (among which all known species produce stalks (*Schilde et al., 2019*)). While there is considerable morphological diversity in fruiting body size and shape among these species (*Raper, 2014*), the chief adaptive function of the fruiting body stalk is thought to be to lift spores into a position where they are more likely to be picked up by a passing arthropod vector. Indeed, one study found that *D. discoideum* fruiting bodies that had been experimentally flattened (reduced to a stalk height of zero) were less likely to be carried by *Drosophila melanogaster* vectors (*Smith, Queller & Strassmann, 2014*). Further, a recent study found that in chimeric fruiting bodies where competing strains reduced their investment to stalk production, fruiting body integrity was compromised and fruiting bodies became significantly more likely to collapse on their own (*Belcher et al., 2022*). Thus, it is likely that the significantly shortened stalks we observed in chimeras containing obligate cheaters would result in reduced dispersal and fitness. Shorter stalks should represent a further limit on obligate cheaters' success in *D. discoideum* populations in nature. The obligate cheater used in this study, EC2, achieved success in the laboratory in an experimental evolution experiment in which fruiting bodies were collected and transferred manually (*Inglis et al., 2017*) and where shorter stalks should therefore not have represented a disadvantage. In nature, where it would have to depend on invertebrate vectors rather than undergraduate ones, we suspect it would not be nearly so successful.

The relationship between the frequency of a cheater within a chimera and stalk height can offer some insight into the mechanism by which cheating is achieved. An obligate *D. discoideum* cheater can become disproportionately represented among the spores of a chimeric cheating body either by being self-promoting, such that it changes its own allocation to produce fewer stalk cells and more spores, or coercive, forcing other lineages to produce more stalk cells and fewer spores (*Buttery et al., 2009*). In a clonal, fully cooperative *D. discoideum* aggregate, approximately 20% of cells are sacrificed to produce

the stalk. If an obligate cheater is fully coercive, we would expect to see full-length stalks produced by chimeric fruiting bodies containing up to about 80% cheaters. Stalk length should only be affected beyond that point, as there would no longer be enough cooperators to produce the required ~20% of stalk cells necessary. Instead, our results show that increasing the frequency of obligate social cheater cells yields shorter fruiting bodies across the entire range. This is more consistent with the cheater using a self-promoting strategy, changing its own allocation without affecting its partner's.

High relatedness is likely sufficient to disincentivize an obligate cheating strategy in *D. discoideum* in nature, but nonetheless *D. discoideum* lineages live and interact in close proximity and opportunities to form chimeric fruiting bodies remain. In reducing the functionality of an important dispersal structure, however, obligate cheaters may experience costs even under low relatedness that make the strategy untenable.

The spread of exploitative cheaters potentially represents a pernicious threat to the long term success of cooperative organisms, and how these threats are mitigated or prevented has long fascinated evolutionary biologists. In some systems, high relatedness imposed by population structure or other mechanisms can make a strategy of obligate cheating untenable. Our results provide evidence for an additional mechanism in the social eukaryote *D. discoideum* that should limit cheater success even when relatedness is lower. Similar mechanisms have been observed in other systems (*Fiegna & Velicer, 2003*; *Dennehy & Turner, 2004*; *Chuang, Rivoire & Leibler, 2010*; *Bruce et al., 2017*). While our experiments were conducted in a laboratory environment, we used a wild-isolated strain and one of its obligate cheater descendants, and demonstrated that even when relatedness is low, obligate cheaters impose a burden on chimeric fruiting bodies that is likely to make them less functional and select against the cheater's success. This additional limit on obligate social cheating in *D. discoideum* paints a more complete picture of why obligate cheaters do not spread in nature and cause the collapse of cooperation.

## ACKNOWLEDGEMENTS

We thank the members of the Queller Strassmann laboratory for helpful discussions and advice.

### Funding

This material is based upon work supported by the National Science Foundation under grant numbers DEB 1753743, and DEB 2237266. The funders had no role in study design, data collection and analysis, decision to publish, or preparation of the manuscript.

### Grant Disclosures

The following grant information was disclosed by the authors:
National Science Foundation: DEB 1753743, and DEB 2237266.

## Competing Interests

The authors declare that they have no competing interests.

## Author Contributions

- James Medina conceived and designed the experiments, performed the experiments, analyzed the data, prepared figures and/or tables, authored or reviewed drafts of the article, and approved the final draft.
- Tyler Larsen analyzed the data, prepared figures and/or tables, authored or reviewed drafts of the article, and approved the final draft.
- David C. Queller conceived and designed the experiments, authored or reviewed drafts of the article, and approved the final draft.
- Joan E. Strassmann conceived and designed the experiments, authored or reviewed drafts of the article, and approved the final draft.

## Data Availability

The data is available at Dryad: Larsen, Tyler; Medina, James; Strassmann, Joan; Queller, David (2024). Data from: In the social amoeba *Dictyostelium discoideum*, shortened stalks may limit obligate cheater success even when exploitable partners are available [Dataset]. Dryad. https://doi.org/10.5061/dryad.5dv41nsd3.

## Supplemental Information

Supplemental information for this article can be found online at http://dx.doi.org/10.7717/peerj.17118#supplemental-information.

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
