# Peer review of "In the social amoeba Dictyostelium discoideum, shortened stalks may limit obligate cheater success even when exploitable partners are available"

_PeerJ, doi:10.7717/peerj.17118_

## Round 0.1 · original submission · Major Revisions

Dear Dr. Medina and colleagues:

Thanks for submitting your manuscript to PeerJ. I have now received three independent reviews of your work, and as you will see, the reviewers raised some minor concerns about the research. Despite this, these reviewers are optimistic about your work and the potential impact it will have on research studying Dictyostelium discoideum and cooperation. Thus, I encourage you to revise your manuscript, accordingly, taking into account all of the concerns raised by both reviewers.

While the concerns of the reviewers are relatively minor, this is a major revision to ensure that the original reviewers have a chance to evaluate your responses to their concerns. There are many suggestions, which I am sure will greatly improve your manuscript once addressed.

Please use the comments by the reviewers to add missing information where possible (all unpublished data as well). Try to restructure your manuscript for clarity, avoiding redundancy and streamlining sections for effective delivery.

Therefore, I am recommending that you revise your manuscript, accordingly, taking into account all of the issues raised by the reviewers. I do believe that your manuscript will be greatly improved once these issues are addressed.

Good luck with your revision,

-joe

Reviewer 1 ·

Basic reporting

no comment

Experimental design

no comment

Validity of the findings

no comment

Additional comments

I enjoyed reading the paper “In the social amoeba Dictyostelium discoideum, shortened
stalks limit obligate cheater success even when exploitable partners are available”. The paper is well-written, and the results are explained clearly and logically. I believe that it also makes a useful contribution to the field. At present, the main explanation for why we don’t find obligate cheaters in nature is because they would sometimes need to aggregate clonally, which they cannot do. This study adds to this, by showing that there is a further cost – obligate cheaters make the stalk less functional.

I have a few comments, all of which are minor

1) In the section on the multiple ways for an amoeba to cheat (lines 67-77), you don’t discuss the second form of “facultative” cheating, where strains will adjust their contribution to the stalk depending on their relative frequency within the chimera (Madgwick 2018 PNAS https://www.pnas.org/doi/10.1073/pnas.1716087115). I appreciate that the focus of this paper is obligate cheating, but it would be good to mention this, as it may be an important aspect of cheating in nature. It is also relevant to your idea of whether cooperative lineages might compensate for the presence of cheaters.
2) Saying relatedness is “not perfect” in nature seems like an odd choice of word. Would “maximal” (or something similar) be better.
3) From each mixture you collected three fruiting bodies at random. Great. Did you then average these? Or are these treated as three independent measures of stalk height?
4) In the results, could you also state the effect size as something like “for every extra 10% of cheaters, stalks were on average Xmm shorter”?
5) Did any of the fruiting bodies collapse? A short fruiting body that has collapsed would have a functional height of 0 (in terms of dispersal). There is some analysis of fruiting body collapse in Madgwick 2018 (same paper as above)
6) Do you think that stalk height is an accurate measure of how many cells are in the stalk? I have read several papers where the authors indirectly infer stalk investment from spore investment (number of starting cells – number of spore cells after aggregation).

Reviewer 2 ·

Basic reporting

This manuscript is well-written in English, and its structure is neat. Additionally, raw data have been provided in an open data repository. The majority of the background information is clearly stated, with appropriate references. However, there are some issues that still need to be addressed:
(1) The title slightly oversells the story. While the authors demonstrate that higher levels of cheaters result in shorter stalks, offering a potential new perspective on the disadvantages of obligate cheaters, they also acknowledge (lines 101-102) that the impact on dispersal potential via the height of fruiting bodies has not been experimentally tested. While it is reasonable to suggest that shortened stalks are likely disadvantageous, as discussed in the manuscript, the title should be revised to better reflect the findings.
(2) The last sentence in lines 44-46 requires supporting references.
(3) The word "versus" should replace "vs" at line 74 to maintain a formal tone.
(4) At line 106, references for the fxbA- strains are necessary.
(5) At line 138, the number "[14]" is unclear and needs clarification.

Experimental design

The experimental design is very clear and demonstrates a distinct result: increased proportions of cheaters lead to a decrease in stalk height. However, two issues need addressing before publication:
(1) When introducing the cheater strain at lines 105-106, it would be beneficial to provide additional information about this evolved strain. What are its characteristics, and why was it chosen? Additionally, the strain named EC2, mentioned in the cited paper, appears to be unclear or undefined [as I could not find a strain named this in the citation (Inglis, Ryu, et al. 2017)].
(2) Given the focus on obligate cheaters in this study, it would be valuable to explore if facultative cheaters exhibit similar behavior, either through previous studies or lab experiments. Furthermore, the rationale for choosing the EC2 obligate cheater strain over the fxbA- strain needs clarification. Are similar results anticipated with different strains?
(3) Although the statistical methods clearly indicate that total spore count does not significantly affect stalk height, including a supplementary figure to illustrate the lack of correlation between these variables could be beneficial.

Validity of the findings

The authors have conducted replications and employed appropriate statistical methods to support the findings of this study. A minor suggestion would be to enhance the figure legend by providing additional details about the data, as currently, the figure only has titles.

Reviewer 3 ·

Basic reporting

This study is a helpful contribution to understanding likely dynamics of obligate cheaters in social amoebae. It is well designed and performed and the paper is nicely written overall. The central finding is a negative correlation between initial cheater frequency in developing groups of Dictyostelium discoideum and final fruiting body height. Under the assumptions that fruiting body height correlates with dispersal rate and that dispersal rate is an important aspect of fitness, the authors conclude that obligate cheaters limit their own spread by reducing fruiting body height as they increase in frequency.

Line 15. It's stated in the abstract that obligate cheaters can lead to population extinction. This point seems significant enough to warrant being addressed further in the main manuscript, but it seems limited to the abstract. Has this been observed in Dictyostelium?

Line 18. Could the failure to find obligate cheaters in nature be due in part to sampling methods rather than actual absence? For example, is D. discoideum normally sampled from tall fruiting bodies? If so, could this bias against detecting obligate cheaters, which, as this study shows, would tend to reduce fruiting-body height? If normal isolation methods don't require development of tall fruiting bodies (or any fruiting bodies at all), however, this would make the failure to find obligate cheaters more striking. I suggest that the authors comment on any possible method bias.

The paper would likely engage a broader readership more deeply if more connections were made to other biological systems in which obligate cheaters have been studied, including quite a few species of bacteria, highlighting commonalities of negative group-level effects of cheaters on cheater fitness. Just one example is Chuang et al 2009 (DOI: 10.1126/science.1166739).

Experimental design

The experimental design and methods section are sound.

Validity of the findings

I don't see that the raw data have been made available.

---

## Round 0.2 · accepted · Accept

Dear Dr. Medina and colleagues:

Thanks for revising your manuscript based on the concerns raised by the reviewers. I now believe that your manuscript is suitable for publication. Congratulations! I look forward to seeing this work in print, and I anticipate it being an important resource for groups studying Dictyostelium discoideum and cooperation.. Thanks again for choosing PeerJ to publish such important work.

Best,

-joe

Reviewer 1 ·

Basic reporting

no comment

Experimental design

no comment

Validity of the findings

no comment

Additional comments

The authors have addressed all of my comments

Reviewer 2 ·

Basic reporting

The authors have comprehensively addressed all the concerns raised in my initial review. The revisions undertaken have significantly improved the manuscript, bringing it to a state that, in my opinion, merits acceptance for publication.

Experimental design

No comment

Validity of the findings

No comment